# Development of an In Vitro Methodology to Assess the Bioequivalence of Orally Disintegrating Tablets Taken without Water

**DOI:** 10.3390/pharmaceutics15092192

**Published:** 2023-08-24

**Authors:** Toshihide Takagi, Takato Masada, Keiko Minami, Makoto Kataoka, Shinji Yamashita

**Affiliations:** 1Faculty of Pharmaceutical Sciences, Setsunan University, Osaka 573-0101, Japan; toshihide.takagi@setsunan.ac.jp (T.T.); takato.masada@setsunan.ac.jp (T.M.); keiko.minami@pharm.setsunan.ac.jp (K.M.); makoto@pharm.setsunan.ac.jp (M.K.); 2Research Organization of Science and Technology, Ritsumeikan University, Kusatsu 525-8577, Japan

**Keywords:** bioequivalence, orally disintegrating tablets, biorelevant dissolution, gastric emptying, gastrointestinal physiology, drug permeation

## Abstract

To assess the probability of bioequivalence (BE) between orally disintegrating tablets (ODTs) taken without water and conventional tablets (CTs) taken with water, an in vitro biorelevant methodology was developed using the BE Checker, which reproduces fluid shifts in the gastrointestinal tract and drug permeation. In addition to the fluid shift from the stomach to the small intestine, the process of ODT disintegration in a small amount of fluid in the oral cavity and the difference in gastric emptying caused by differences in water intake were incorporated into the evaluation protocol. Assuming a longer time to maximum plasma concentration after oral administration of ODTs taken without water than for CTs taken with water due to a delay in gastric emptying, the fluid shift in the donor chamber of the BE Checker without water was set longer than that taken with water. In the case of naftopidil ODTs and CTs, the values of the f_2_ function, representing the similarity of the permeation profiles, were 50 or higher when the fluid shift in ODTs taken without water was set at 1.5 or 2 times longer than that of the CTs taken with water. The values of the f2 function in permeation profiles of pitavastatin and memantine ODTs were both 62 when the optimized experimental settings for naftopidil formulations were applied. This methodology can be useful in formulation studies for estimating the BE probability between ODTs and CTs.

## 1. Introduction

Oral dosage forms are widely used, as they can be taken without the assistance of medical personnel. For the clinical development and the first launch of oral drugs with new molecular entities, they are usually formulated as conventional tablets (CTs) or capsules because the pharmaceutical industry has extensive experience in the design and manufacturing of CTs. These can later be switched or replaced by orally disintegrating tablets (ODTs) with the increased medical needs. After the exclusivity of the drugs expires, generic drug companies may newly develop ODTs to differentiate their products from the original medicines or those from other generic companies. In the formulation design of ODTs, they need to disintegrate rapidly when placed on the tongue with a small amount of fluid in the oral cavity, without taking a glass of water. Additionally, they should not have an unpleasant taste or mouth feel when taken without water. There are also challenges in formulating ODTs in mechanical strength, hygroscopicity, and size of the products because they tend to be fragile, porous, and bulky to give the required formulation properties and mask the taste of any unpalatable components.

Bioequivalence (BE) studies compare the bioavailability (BA) of two drug formulations to ensure the therapeutic equivalence of newly developed drug formulations from those that have confirmed clinical efficacy and safety in human studies. If there are no significant differences in the extent and the rate of BA between them, these formulations are considered to be BE. In the regulatory draft guidance of the U.S. Food and Drug Administration, if the labeling states that the ODTs may be administered with or without water, the BE studies should be conducted without water [1]. In the draft guideline released by the International Council for Harmonisation of Technical Requirements for Pharmaceuticals for Human Use (ICH) on December 2022, if the labeling states that the ODT can be taken with or without water, a three-arm BE study is recommended to determine the BE of the ODT administered with and without water compared to the comparator product [2].

Before conducting in vivo BE studies in healthy volunteers, the drug dissolutions from the formulations are evaluated by an in vitro compendial dissolution apparatus to confirm their characteristics. Those compendial dissolution methodologies conducted with fixed conditions in terms of pH, volumes, and fluid components are useful for the quality assurance of drug products. However, because they do not reflect dynamic gastrointestinal (GI) processes, such as the transit of the dissolved drugs or formulations and the fluid shift from the stomach to the small intestine, and bile acid secretion into the small intestine, the in vivo performance of the drug formulations with pH-dependent dissolution may not be evaluated precisely. Weakly basic drugs often supersaturate in neutral pH after the fluid shift from acidic conditions, and bile acids in the small intestine significantly promote drug dissolution. In addition, the excipients in the formulation often affect the behavior of supersaturation and dissolution of the drugs. To overcome these limitations, biorelevant dissolution methodologies—such as a two-stage, single-compartment model [3,4,5,6]; a transfer model [7,8]; a gastrointestinal simulator [9,10,11,12,13,14,15,16,17]; and an artificial stomach–duodenum model [18,19]—have been developed. Except for a single-compartment model, these models have two or three compartments representing the dynamic physiological changes from the stomach to the small intestine. Pumps are used to transfer fluids between the compartments. They have successfully simulated the performance of formulations in the GI tract, but suffer from complex setups and low system throughput. In our previous study, we developed an in vitro bioequivalence checking system (BE Checker), which reproduces the fluid shift from the stomach to the small intestine by simply adding concentrated simulated intestinal fluid into the simulated gastric fluid in a single chamber [20]. In addition, the membrane permeation of the dissolved drugs through the hydrophilic filter to the receiver chamber filled with octanol can be evaluated simultaneously. We have applied the BE Checker to assess the formulation performance of oral drug products and demonstrated that it enabled the assessment of drug dissolution and permeation profiles from the formulations under various GI conditions by changing the pH, fluid composition, infusion rate, and stirring rate, reflecting those in healthy and special populations. An approved naftopidil CT and ODT (Flivas^®^) were used as standard drug products, and the experimental conditions for the administration with a glass of water were optimized. In the case of Flivas^®^ ODT, disintegration in the administered water was necessary to avoid the dissolution delay by forming a gel-like layer on the surface of the ODTs when they were administered directly into the acidic simulated gastric fluid.

Since ODTs are taken with or without a glass of water, their BA taken without water needs to be BE with that of the comparator products. When the dosage form of the comparator products is CTs taken with water, ODTs have to be designed to exhibit BE with the CTs, even if ODTs are administered without water. In the case of CTs taken with water, the administered water increases the initial fluid volume in the stomach, but for the ODTs taken without water, they need to be dissolved in about 30 mL of the resting fluid volume in the stomach [21]. Although the fluid amount in the GI tract significantly impacts the dissolution of the drugs from the formulations, in vitro methodologies to assess BE do not consider the difference in the fluid volume when ODTs are taken without water. Moreover, gastric emptying is dependent on the amount of administered water [22]. Even though the transit time of the drugs and the formulations in the GI tract are necessary to be considered for BE prediction, no methodologies have been reported to the best of our knowledge.

In this study, we developed the BE Checker to assess the BE of ODTs taken without water by customizing the size of the donor chamber suitable to control for gastric fluid volume. In addition, by incorporating the process of ODT disintegration in a small amount of fluid in the oral cavity and the difference in gastric emptying caused by water intake in the evaluation protocol, the dissolution and permeation of the drugs from ODTs taken without water were evaluated. By evaluating the dissolution and permeation of the pitavastatin and memantine formulations using the optimized methodology for the naftopidil formulations, we confirmed that this methodology is applicable to drugs with other physical properties. Thus, we developed an in vitro methodology to assess the BE between CTs and ODTs from the similarity of the permeation profiles evaluated in the BE Checker.

## 2. Materials and Methods

### 2.1. Materials

Hank’s balanced salt solution was obtained from Gibco Laboratories (Lenexa, KS, USA). Egg phosphatidylcholine was obtained from Fujifilm Wako Pure Chemical (Osaka, Japan). Sodium taurocholate was obtained from Nacalai Tesque (Kyoto, Japan). Flivas^®^ tablet 25 mg and Flivas^®^ ODT 25 mg (naftopidil) were obtained from Asahi Kasei Pharma (Tokyo, Japan). Livalo^®^ tablet 1 mg and Livalo^®^ ODT 1 mg (pitavastatin calcium hydrate) were obtained from Kowa Company (Aichi, Japan). Memary^®^ tablet 5 mg and Memary^®^ ODT 5 mg (memantine hydrochloride) were obtained from Daiichi Sankyo (Tokyo, Japan). All other reagents used were of the highest grade available.

### 2.2. Design of the BE Checker

The BE Checker was developed to assess the dissolution of the drugs from the formulations and their membrane permeation by reproducing the environmental changes in the GI tract by altering the pH, composition, and volume of the fluid in a donor chamber and simultaneously assessing drug transport through the hydrophilic filter into the octanol in a receiver chamber (Figure 1A) [20]. A hydrophilic filter with a pore size of 0.22 µm (Durapore^®^, Merck Millipore, Burlington, MA, USA) was mounted between the chambers. The effective surface area of the filter was 6.60 cm^2^. Both chambers were made of acrylic plastic. For the evaluation of the CTs taken with water, the fluid volume of the donor chamber was 40 mL in the bottom, which corresponds to the stomach, and 100 mL in total (Figure 1A). As the BE Checker was designed as a 1/4 scale of the human GI tract in volume, 40 mL was set for the volume of the bottom of the donor chamber. This is about 1/4 of the total volume of 150 mL, the recommended volume of the administered water in BE studies in Japan [21], and 30 mL of the resting volume in the stomach. Assuming that the total volume of the fluid in the small intestine to which drugs were exposed was about 400 mL, the final volume in the donor chamber was set to 100 mL. For the evaluation of ODTs taken without water, the fluid volume of the donor chamber was 5 mL in the bottom and 65 mL in total (Figure 1B).

In the case of the evaluation of CTs taken with water, the CT was administered into 37.5 mL of distilled water at the bottom of the donor chamber, representing the CT contacting water in the mouth. After 1 min of incubation, 2.5 mL of 16-times concentrated FaSSGF (pre-FaSSGF) was added to the donor chamber to shift the donor fluid to the gastric conditions. After an additional 1 min incubation, 1.67-times concentrated FaSSIF (pre-FaSSIF) was infused into the donor chamber at a rate of 6.0 mL/min for 10 min from the side inlet to shift the fluid to the intestinal conditions. In our previous article, we confirmed that the infusion rates of 6.0 mL/min for 10 min and 3.0 mL/min for 20 min were both appropriate for the evaluations of naftopidil CT and ODT taken with water when the stirring rate was 100 rpm [20]. Experimental conditions in the BE Checker for ODTs taken without water representing the human GI physiology are shown in Figure 2. ODT was administered into 2.5 mL of distilled water at the bottom of the donor chamber. After 1 min of incubation, 2.5 mL of twice-concentrated FaSSGF was added to the donor chamber. After an additional 1 min of incubation, 1.08-times concentrated pre-FaSSIF was infused into the donor chamber from the side inlet to shift the fluid to the intestinal conditions. The infusion rate of the pre-FaSSIF was changed depending on the experimental design: 6.0 mL/min for 10 min, 4.5 mL/min for 15 min, and 3.0 mL/min for 20 min. The initial pH of the donor fluid after adding pre-FaSSGF was 1.6, and the final pH of the donor fluid after the pre-FaSSIF infusion was 6.5. The details of the compositions of the donor fluids were described in our previous article [20]. The receiver chamber was empty at the beginning, and octanol was then infused into the receiver chamber at a rate that maintained the height of the fluid surface the same as that of the donor chamber. The time profiles of the results were expressed after adding pre-FaSSGF into the donor chamber. All the donor fluids were prewarmed to 37 °C. The BE Checker was set up on a constant temperature plate at 37 °C and covered with an insulation board during the experiments.

Samples were taken from each of the chambers at the specified time. Donor samples were filtered through a membrane filter (Millex^®^-LH, pore size: 0.45 µm, Merck Millipore, Burlington, MA, USA) to remove the undissolved and precipitated drug. The volume of each side was maintained by adding fresh fluid after sampling.

### 2.3. Analytical Methods

Drug concentrations in the samples were determined using an ultra-performance liquid chromatography (UPLC) system (Acquity^®^ UPLC, Waters, MA, USA) equipped with a tandem mass spectrometer (Acquity^®^ TQD, Waters, MA, USA). A reversed-phase Waters Acquity UPLC BEH C18 analytical column (50 mm × 2.1 mm, 1.7 µm particle size) was used, with a mobile phase consisting of 0.1% formic acid (solvent A) and acetonitrile containing 0.1% formic acid (solvent B). The initial mobile phase was 98% solvent A and 2% solvent B, pumped at a 0.3 mL/min flow rate. The percentage of solvent B increased linearly to 95% for 1.0 min and was then maintained for 1.0 min. Between 2.01 and 2.5 min, the percentage of solvent B decreased linearly to 2% and was maintained until the end of the run time of 3.0 min. All samples were injected at 5 µL into the UPLC system. The molecular mass of the analytes was monitored in positive ionization mode. For naftopidil, the *m*/*z* values of the precursor and production ions were 393.39 and 190.09, respectively. For pitavastatin, they were 423.26 and 275.22, respectively. For memantine, they were 180.18 and 163.18, respectively.

### 2.4. Simulated pH–Time Profiles in the Donor Chamber of the BE Checker

For the evaluation of the fluid pH shift in the donor chamber for CTs taken with water, it was monitored every minute by infusing 60 mL of 1.67-times concentrated pre-FaSSIF into 40 mL of FaSSGF at a rate of 6.0 mL/min for 10 min. For the evaluation of ODTs taken without water, it was monitored every minute by infusing 60 mL of 1.08-times concentrated pre-FaSSIF into 5 mL of FaSSGF at a rate of 6.0 mL/min for 10 min. As there may be a delay in gastric emptying of ODTs taken without water, pH–time profiles of the donor chamber for ODTs at a rate of 4.5 mL/min for 15 min and 3.0 mL/min for 20 min were also evaluated.

### 2.5. The Similarity of the Permeation Profiles between CTs and ODTs

To assess the similarity of the permeation profiles between CTs and ODTs, the values of the f_2_ function were calculated using the following Equation (1):(1)$$f_{2}=50\times log\left[\frac{100}{\sqrt{1+\frac{\sum_{i=1}^{n}{\left(T_{i}-R_{i}\right)}^{2}}{n}}}\right]$$

T_i_ and R_i_ indicate the permeated amount (% of dose) in the test (ODT) and reference (CT) formulation, respectively, and n indicates the number of measurement points to be compared. In the calculation, the f_2_ was evaluated from four data points, 1/4T_a_, 1/2T_a_, 3/4T_a_, and T_a_, where the permeated amount from the standard tablet at 120 min was set as 100%, and T_a_ was the time point of about 85%. When the value of the f_2_ was above 50, which indicates a 10% difference overall, the permeation profiles were considered similar.

## 3. Results

### 3.1. Experimental Settings for the Evaluation of Disintegration/Dissolution and Permeation of ODTs Taken without Water in the BE Checker

When ODTs are taken without water, their disintegration and drug dissolution occur in a small volume of fluid in the mouth and the stomach compared with the volume taken with a glass of water. In our previous studies on the development of the BE Checker for the evaluation of the formulations taken with water [20], the bottom of the donor chambers was 40 mL, about one-quarter of the human gastric fluid volume after drug administration with 150 mL of water, a recommended volume for BE studies in Japan [23]. For the evaluation of ODTs taken without water, considering the difference in the fluid volume and usability in the experiments, a donor chamber with 5 mL in the bottom was designed and used throughout the studies (Figure 1B). In the preliminary evaluations of donor chambers with 3 or 10 mL in the bottom, 3 mL was not feasible for precise mixing by the paddle, and 10 mL was considered larger than the appropriate fluid volume.

In addition to the difference in fluid volumes, the difference in the gastric emptying time was considered. Table 1 summarizes the literature information on the time to maximum plasma concentration (T_max_) after oral administration of ODTs taken with or without water [24,25,26,27,28]. The T_max_ of ODTs taken without water tended to be longer than that with water, and the average ratio was 1.6. Assuming that the difference in T_max_ was derived from gastric emptying time, the infusion time of the pre-FaSSIF into the donor chamber in ODT tests was set at 1.5 or 2 times longer than that of the CTs taken with water. Figure 3 shows the simulated pH–time profiles of the donor fluid in the BE Checker when the initial gastric pH was 1.6 and the final intestinal pH was 6.5. Because the fluid volume in the bottom was 5 mL for the ODT evaluations, the pH shift was faster than in the CT evaluations (40 mL in the bottom) when 60 mL of pre-FaSSIF was infused into the donor chamber.

### 3.2. Disintegration/Dissolution and Permeation of Naftopidil CTs Taken with Water and ODTs Taken without Water in the BE Checker

Naftopidil CTs and ODTs were used to assess the similarities of the disintegration/dissolution and permeation of the drugs between CTs taken with water and ODTs taken without water. Naftopidil has a basic pKa of 3.7, with a high solubility at acidic pH levels and low solubility at neutral pH. The formulations used in this study confirmed their BE in humans when administered in fasted conditions [24]. Figure 4 shows the dissolution and permeation profiles of naftopidil from CTs and ODTs in the BE Checker when the stirring rate was 50 rpm. The results of CTs evaluated when administered with water in the BE Checker were referenced from our previous article [20]. In the early periods, drug dissolution from CTs was delayed, and the dissolved drug amount and concentrations from ODTs were higher than those from CTs, regardless of the infusion time of the pre-FaSSIF, reflecting their formulation characteristic of very rapid disintegration. Also, the dissolved drug concentrations in the 10 min infusion of the pre-FaSSIF were lower than those in 15 min infusion, showing that the rapid pH shift to neutral conditions in the 10 min infusion affected the solubility of weakly basic compounds. Although the dissolved drug concentrations of CTs and ODTs were comparable after the fluid conditions were the same after infusion was completed, the permeated drug profiles in the receiver sides were not similar between CTs and ODTs, even though they were BE in humans. Considering that the stirring rate of 50 rpm did not reflect the agitation of the fluid in the human GI tract, we tried to set a new condition for more precise evaluations.

Figure 5 shows the dissolution and permeation profiles of naftopidil from CTs and ODTs in the BE Checker when the stirring rate was 100 rpm. In contrast to the results at 50 rpm, the dissolved drug amount from CTs initially increased to 67%, then decreased after the beginning of pre-FaSSIF infusion (Figure 5A). The dissolved drug concentration from CTs was also improved in the early period (Figure 5B). Corresponding with the increased dissolution of the drug from CTs, the permeated drug profiles in the receiver side between CTs and ODTs were comparable when the infusion time of pre-FaSSIF into the donor chamber in ODTs was set at 1.5 or 2 times longer than that of CTs. When the drug permeations were compared without considering the difference in gastric emptying time, which corresponds to the infusion time in the BE Checker, the permeated drug profile of ODTs with the 10 min pre-FaSSIF infusion was lower than those in CTs with 10 min pre-FaSSIF infusion in the late period (Figure 5C).

To quantitatively evaluate the similarity of the drug permeation profiles, the values of the f_2_ function between CTs and ODTs were calculated and are summarized in Table 2. If the drug permeation profiles are identical or 10% different, the f_2_ values are 100 or 50, respectively [29]. At a stirring rate of 50 rpm, the f_2_ values were minimal, even if the difference in the gastric emptying time was considered. On the other hand, when the dissolution and permeation were evaluated at 100 rpm while considering the difference in the gastric emptying time, the values of the f_2_ function were greater than 50. These results suggested the importance of incorporating the difference in gastric emptying time between CTs taken with water and ODTs taken without water in evaluations by the BE Checker.

### 3.3. Validation Studies Using Other CTs and ODTs in the BE Checker

#### 3.3.1. Pitavastatin

Pitavastatin formulations were used in the validation studies to confirm the adequacy of the optimized experimental setups by the naftopidil formulations. Because the average ODT T_max_ ratio in clinical studies was 1.6 (Table 1), the validation studies for ODTs were performed with 15 min of pre-FaSSIF infusion. Pitavastatin has an acidic pKa of 4.4 [30]. Figure 6 shows the dissolution and permeation profiles of pitavastatin from CTs and ODTs in the BE Checker when the stirring rate was 100 rpm. Reflecting the difference in disintegration and dissolution characteristics from the formulations, the drug dissolutions from the ODTs were completed even at the first time point. In contrast, drug dissolution from the CTs gradually increased in the early time points. The drug dissolution was completed after the fluid shift from the gastric to the small intestinal conditions (Figure 6A). The dissolved drug concentrations from ODTs were higher than those from CTs because of the difference in the total volume in the donor chambers (65 mL for ODTs or 100 mL for CTs). Regarding the drug permeation profiles, when the infusion time of pre-FaSSIF was 10 min in ODTs and CTs, the drug permeation from ODTs was higher than that from CTs, and the value of the f_2_ function was 44. On the other hand, when the infusion time was 15 min in ODTs, which was 1.5 times longer than in CTs, the drug permeations from CTs and ODTs were similar, and the value of the f_2_ function was 62 (Figure 6C). These results indicated that the combination of the stirring rate and infusion time of pre-FaSSIF was essential for evaluating BE and that the experimental settings used in this study were appropriate.

#### 3.3.2. Memantine

Another validation study was conducted using memantine formulations. Memantine has a basic pKa of 10.6; thus, the dissolved drugs are protonated in physiological GI fluids. Its solubility between pH 1 and 9 is 30 mg/mL or higher. Figure 7 shows the dissolution and permeation profiles of memantine from CTs and ODTs in the BE Checker when the stirring rate was 100 rpm. The dissolution profiles of the memantine formulations were similar to those of the pitavastatin formulations. That is, the drug dissolution from CTs was slower than that from ODTs in the early period, and the drugs were well dissolved after the fluid shift from the gastric to the intestinal conditions (Figure 7A,B). In contrast, the drug permeation profiles from CTs and ODTs were similar, regardless of the infusion time of pre-FaSSIF (Figure 7C). The values of the f_2_ function at 10 min and 15 min of infusion were 58 and 62, respectively.

## 4. Discussion

We demonstrated that the BE Checker with the customized size of the donor chamber with 5 mL in the bottom could be a promising in vitro system to assess the BE of ODTs taken without water. In the method development, to incorporate the process where ODTs contact saliva, the ODT was administered into 2.5 mL of distilled water, and after 1 min of incubation, 2.5 mL of twice-concentrated FaSSGF was added to the donor chamber to shift the fluid conditions from those of the oral cavity to those of the stomach. This process was critical to reproduce disintegration in the oral cavity and the transfer to the stomach. As we demonstrated in the previous article, the disintegration of naftopidil ODT was disturbed by the formation of the gel-like layer on the surface when it was administered directly into FaSSGF [20]. Considering the setting validity and the simplicity of the protocols for ODTs taken without water and CTs taken with water, the same volume (2.5 mL), but different concentrations of FaSSGF were added to the fluids where formulations were administered. The volume of human saliva in the oral cavity is about 1.1 mL before swallowing [31], and the secretion of saliva at 0.3 to 0.4 mL/min under normal conditions or 1.5 to 2.0 mL/min under stimulated conditions continuously flows to the stomach through the esophagus [32]. Assuming that the amount of saliva after taking ODTs is increased due to stimulation, the initial water volume of 2.5 mL in the bottom of the donor chamber for ODTs taken without water can precisely represent the in vivo situation.

As summarized in Table 1, the average ratio of T_max_ of ODTs taken without water was 1.6 times longer than that of ODTs taken with water. By adopting 1.5 or 2 times longer infusion time of the pre-FaSSIF into the donor chamber, we obtained similar permeation profiles between naftopidil ODT taken without water and CT with water (Figure 5). In this condition, the value of the f_2_ function was above 50, which means that there was a less than 10% difference between the profiles (Table 2). It was reported that gastric emptying depended on the administered water volume. The resting fluid volume in the stomach was about 30 mL, and the gastric emptying rate was about 2 mL/min [21]. When large volumes of water were ingested, the initial gastric emptying rate was higher (10–40 mL/min) [33]. Grimm et al. reported that the gastric emptying profiles after 240 mL of water were good first-order kinetic, and the gastric half-emptying time was about 15 min. In contrast, when 20 mL of water was administered, the profiles were zero-order on average with large variability, and the half-emptying time was about 45 min [22]. The delayed gastric emptying results in a longer exposure of the drugs to low pH in the stomach, which increases the dissolution rate for basic drugs. As shown in Figure 5, the dissolved concentrations of naftopidil from ODTs taken without water were increased by changing the infusion time of pre-FaSSIF to 15 or 20 min, resulting in improved permeation profiles compared with the results obtained via 10 min infusion. From a pharmacokinetic point of view, T_max_ is when the absorption rate into the body and the elimination rate from the body are balanced. Therefore, we referred to the average ratio of T_max_ of ODTs taken with or without water for optimizing the infusion time of pre-FaSSIF into the donor chamber.

Dissolved drug concentrations in the donor fluids also can be a crucial determinant for BE between CTs and ODTs because the drug absorption rate depends on the dissolved drug concentration in the small intestine and the gastric emptying rate. In the case of CTs taken with 150 mL of water, the drug was initially dissolved in approximately 180 mL of fluid in the stomach (a resting 30 mL of gastric fluid plus 150 mL of water). The dissolved drugs and disintegrated particles of the formulations were transferred with the gastric fluid to the small intestine through the pylorus. In contrast, dissolved drug concentrations in the stomach from ODTs taken without water should be higher because of their rapid disintegration and a small volume of gastric fluid (about 30 mL). However, considering the slow gastric emptying when ODTs were taken without water, it might compensate for the higher dissolved drug concentration in the gastric fluid to be BE with CTs taken with water.

When the stirring rate was 50 rpm, the permeation profiles between naftopidil CTs and ODTs were separated, even in the prolonged infusion time in ODT evaluations (Figure 4). When dissolved drug concentrations from CTs at early time points were compared between 50 and 100 rpm (Figure 4A and Figure 5A), they were lower at 50 rpm than at 100 rpm. In our previous studies for BE assessments between CTs and ODTs taken with water by the BE Checker, the stirring rate of 50 rpm was not optimal, but 100 rpm was appropriate [20]. These suggested that 50 rpm for CTs did not reflect the agitation strength in the stomach precisely. Based on the theory of fluid dynamics, agitation strength is proportional to the cube of the stirring rate [34], which means eight times stronger agitation was achieved at 100 rpm. Because the mixing of the fluid in the human GI tract occurs mainly by periodic peristaltic movement, it is difficult to say the stirring rate of 100 rpm in the BE Checker is always biorelevant. Considering similar permeation profiles between tablets and ODTs were obtained at 100 rpm, we thought the stirring rate of 100 rpm in the BE Checker was applicable for validation studies using pitavastatin and memantine formulations.

Although the drug permeations from ODTs depended on the pre-FaSSIF infusion time, the trends differed between naftopidil and pitavastatin. For naftopidil, the permeation profiles were higher in prolonged infusion settings that were longer than 10 min (Figure 5C), but for pitavastatin, the results were the opposite (Figure 6C). As shown in Figure 3, the pH of the donor fluids shifted rapidly when pre-FaSSIF was infused in a short period of time. In the case of naftopidil, because it is a weakly basic drug, its solubility is high in the low pH range. Therefore, the drug dissolution and permeation were delayed when the fluid pH shifted rapidly to neutral in 10 min of pre-FaSSIF infusion. On the other hand, because pitavastatin has an acidic pKa of 4.4, the effect of the pH shift on the drug dissolution and permeation was the opposite. It was suggested that the combination of 100 rpm stirring rate and 15 min pre-FaSSIF infusion worked for both acidic and basic drug formulations. The drug permeation profiles of memantine ODTs were similar, regardless of the infusion time of pre-FaSSIF (Figure 7). Memantine is a typical BCS class III compound with a high solubility and low permeability. Memantine was kept protonated because its basic pKa of 10.6 is higher than the physiological GI pH range. Also, a solubility of 30 mg/mL is high enough to dissolve 5 mg of the drug in the formulation; its dissolution was not dependent on the infusion time of the pre-FaSSIF. Therefore, drug permeation was similar in both infusion time settings.

It is possible the drug permeation in the BE Checker may be affected by the excipients, depending on their type or amounts in the formulation. Although the information on some of the excipients used in the formulations is disclosed, their amounts are not available for the formulations on the market. We are planning to evaluate the dissolution and permeation of the drug formulations made with different amounts of excipients in future experiments.

The similarity of the permeation profiles between CTs taken with water and ODTs taken without water was evaluated using the f_2_ function. This is because it is applicable to basic drugs, like naftopidil, in which the permeation profiles were not linear due to the decrease in dissolution after the fluid shift to FaSSIF. However, it is necessary to verify which time points should be used and how the reference value of the f_2_ function should be set. In the future, we will evaluate oral formulations that failed to show BE in human studies and establish more appropriate criteria with the cooperation of pharmaceutical companies.

As we demonstrated in the studies of dissolution/permeation systems [35,36], we are constructing correlations between the permeated amount in the BE Checker and the fraction absorbed in humans. In addition, in vitro–in silico–in vivo extrapolation (IVISIVE) analysis can also be applied to estimate the AUC or C_max_ ratio and variability between the test and the reference formulations from their permeation profiles in the BE Checker. By further improving the accuracy and precision of our evaluation system and promoting evaluations of formulations including non-BE formulations in the BE Checker, a precise mathematical model for IVISIVE analysis can be constructed.

There are some advantages and disadvantages for all the biorelevant dissolution systems, including the BE Checker. In multiple-compartment systems, when the gastric fluid, where the drug formulation is administered, is transferred to the duodenum compartment by a peristaltic pump, undissolved drug particles may not be precisely transferred. In contrast, the BE Checker has the advantage that the dissolution and permeation of the drugs from the formulation can be assessed simultaneously by simply adding the concentrated simulated intestinal fluid into the simulated gastric fluid in the donor chamber. Also, as the evaluations in the BE Checker have relatively improved throughput, we can carry out three experiments at the same time using a dissolution apparatus with six paddles. One of the disadvantages of the BE Checker is that we cannot switch the donor fluid and can only add concentrated fluids to it. It is necessary to select a method that is suitable for the purpose.

In this study, we applied the same infusion times for naftopidil, pitavastatin, and memantine ODTs taken without water. However, setting an appropriate infusion time for each formulation may be more appropriate when using the BE Checker for formulation studies. Evaluations using gastric motility models, which precisely mimic GI physiology with or without water, can be the right approach for estimating a transfer from the stomach to the small intestine for individual formulations. By incorporating the obtained in vitro parameters of gastric emptying rate in the estimation for T_max_ shift and an appropriate infusion time in the BE Checker, the methodology for evaluating BE between ODTs and CTs will be improved and individualized for each formulation.

## 5. Conclusions

We developed an in vitro methodology to assess the BE between CTs taken with water and ODTs taken without water using the BE Checker with 5 mL of fluid at the bottom of the donor chamber. In addition to the fluid shift from the stomach to the small intestine, by incorporating the process of ODT disintegration in a small amount of fluid in the oral cavity and the difference in gastric emptying caused by differences in water intake in the evaluation protocol, the dissolution and permeation of the drugs from ODTs taken without water and those from CTs taken with water were precisely evaluated for BE prediction. This system can be useful in formulation studies for estimating the BE probability between ODTs and CTs.

## Figures and Tables

**Figure 1 pharmaceutics-15-02192-f001:**
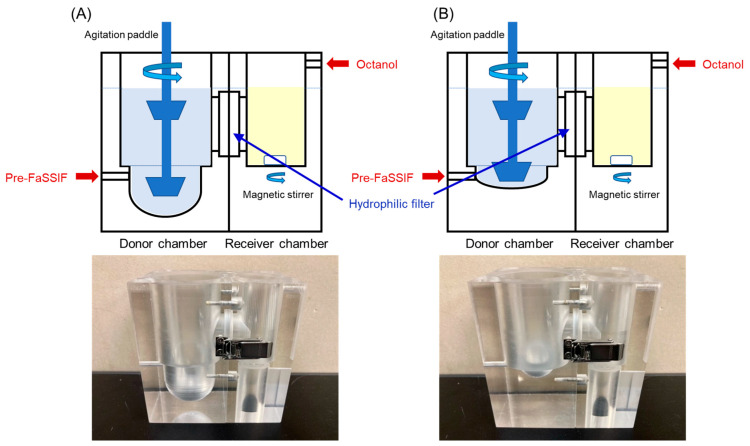
Schematic illustration and pictures of the BE Checker to assess the BE of (**A**) CTs taken with water and (**B**) ODTs taken without water.

**Figure 2 pharmaceutics-15-02192-f002:**
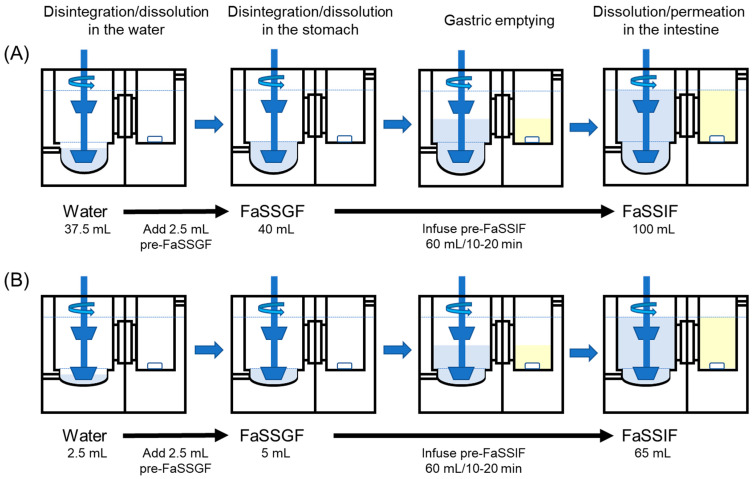
Experimental conditions in the BE Checker for tablets taken with and without water. For (**A**) CTs taken with water, (**B**) ODTs taken without water.

**Figure 3 pharmaceutics-15-02192-f003:**
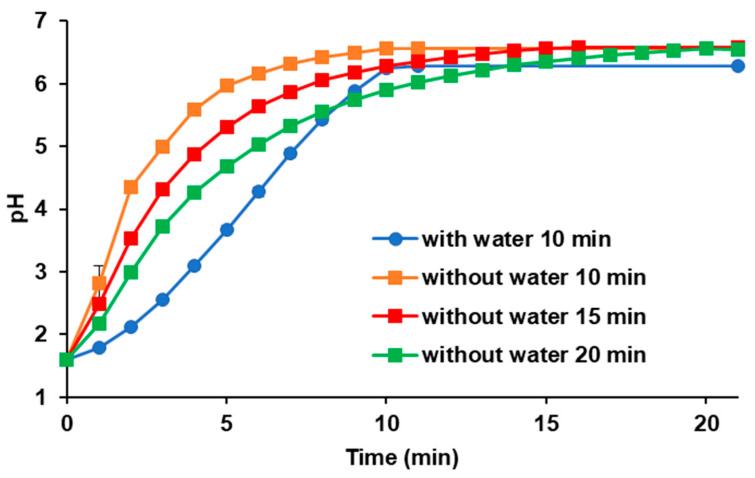
Simulated pH–time profiles of the donor fluid in the BE Checker.

**Figure 4 pharmaceutics-15-02192-f004:**
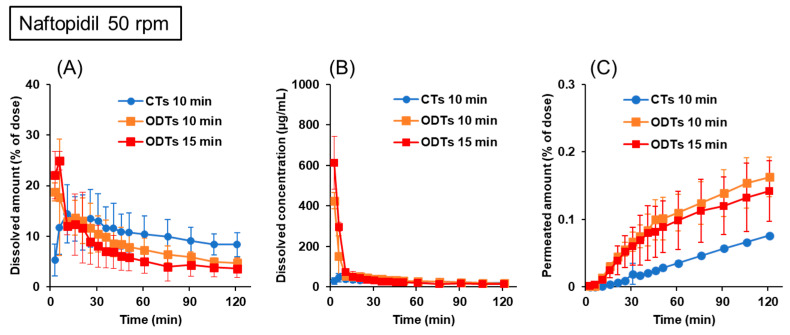
Dissolution and permeation profiles of naftopidil from 25 mg CTs and ODTs in the BE Checker when the stirring rate was 50 rpm. (**A**) Dissolved amount in the donor chamber; (**B**) dissolved concentration in the donor chamber; (**C**) permeated amount in the receiver chamber.

**Figure 5 pharmaceutics-15-02192-f005:**
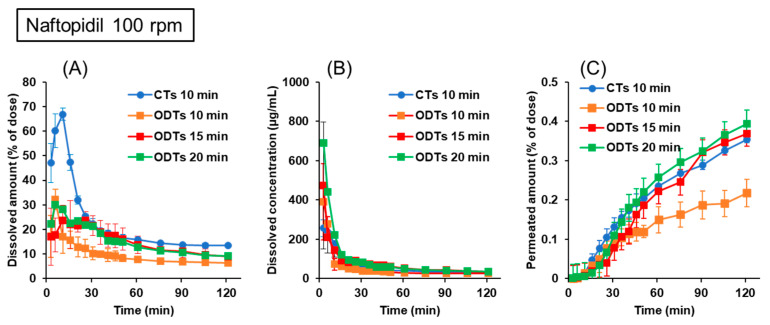
Dissolution and permeation profiles of naftopidil from 25 mg CTs and ODTs in the BE Checker when the stirring rate was 100 rpm. (**A**) Dissolved amount in the donor chamber; (**B**) dissolved concentration in the donor chamber; (**C**) permeated amount in the receiver chamber.

**Figure 6 pharmaceutics-15-02192-f006:**
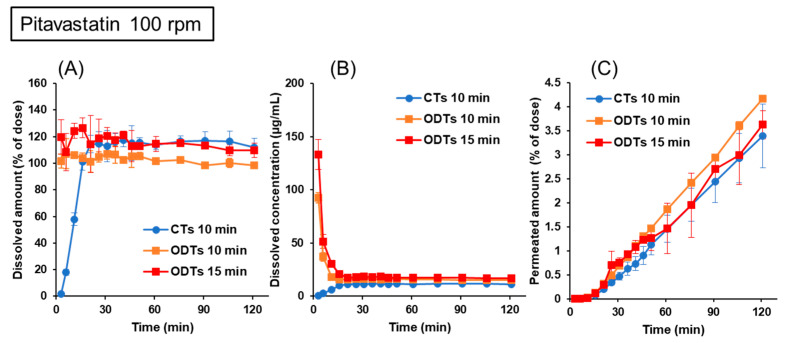
Dissolution and permeation profiles of pitavastatin from 1 mg CTs and ODTs in the BE Checker when the stirring rate was 100 rpm. (**A**) Dissolved amount in the donor chamber; (**B**) dissolved concentration in the donor chamber; (**C**) permeated amount in the receiver chamber.

**Figure 7 pharmaceutics-15-02192-f007:**
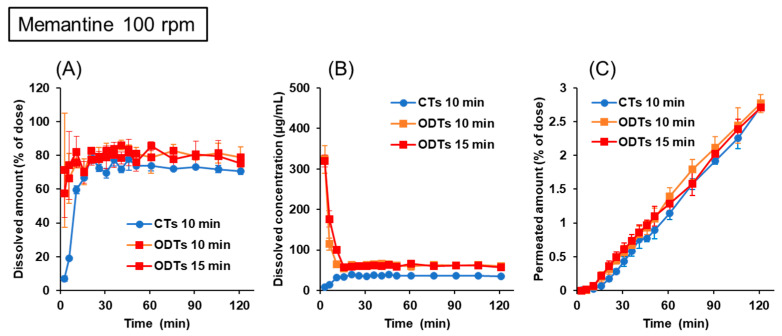
Dissolution and permeation profiles of memantine from 5 mg CTs and ODTs in the BE Checker when the stirring rate was 100 rpm. (**A**) Dissolved amount in the donor chamber; (**B**) dissolved concentration in the donor chamber; (**C**) permeated amount in the receiver chamber.

**Table 1 pharmaceutics-15-02192-t001:** Comparison of T_max_ after oral administration of ODTs with or without water.

	ODT T_max_ (h)		Ratio
	with Water	without Water	
Naftopidil	0.5	0.8	1.6
Silodosin	1.04	1.46	1.4
Pregabalin	0.5	1	2.0
Imidafenacin	1.0	1.4	1.4
Average			1.6

**Table 2 pharmaceutics-15-02192-t002:** Values of the f_2_ function in drug permeation profiles between naftopidil CTs and ODTs.

Pre-FaSSIF Infusion (min)		Stirring Rate (rpm)	Values of the f_2_ Function
CTs	ODTs		
10	10	50	1.5
	15		6.3
	10	100	32
	15		50
	20		54

## Data Availability

The data presented in this study are available in this article.

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
