# Peer review of "Development of an In Vitro Methodology to Assess the Bioequivalence of Orally Disintegrating Tablets Taken without Water"

_pharmaceutics, 2023, doi:10.3390/pharmaceutics15092192_

Round 1
Reviewer 1 Report
Just one comment. Since in vivo profiles have been published in some of the cited literature, why did not the authors attempt to establish some kind of an IVIVC?
Reviewer 2 Report
Dear Authors,
Presented work is interesting and have great potential.
I have several questions:
Abstarct should have more informations conected with resalts. It is too general and does not convay idea of work
Introduction
Physiologocal base for BE checker....or some investigations conected to what is going on in human organism when taken with or without water is missing
More about selected API should be written and why they are interesting
Methodology
Design of BE checker: Wy these specific amoints of liquids are used. Is there smoe experimental design that was proven these are amounts that should be used.
Simulation of pH profile: againe rate of medium change in ml/min is based on what knowledge
Analytics: is F2 juistified in these noncompendial diss. Could some other statistic be more useful
Results
Fig 3: time points with SD should be placed- not just lines
Why time points 10, 15 and 20 minites are choosen
Since Tmax shift when taken with or without water is it conected to physiology change. Water usully leabe stomach very fast. More explanation how these is conected to dissolution should be written in tesults and discussion.
Best of luck
Reviewer 3 Report
The manuscript entitled "Development of an In Vitro Methodology to Assess the Bioequivalence of Orally Disintegrating Tablets Taken Without Water", submitted by Toshihide Takagi et al, continues the testing and fine tuning of the authors' previously introduced "BE Checker". The present manuscript deals with the thoroughly investigated BE of naftopidil ODT taken without water and CT taken with water. With the optimized experimental parameters thus obtained, the BE Checker is successfully tested in the BE assessment of pitavastatin and memantine ODTs. The permeation profiles presented call for some (simplified) kinetic analysis that could offer a better BE assessment tool than the utilized f2 function. My opinion is that in future developments permeation kinetics would bring sizable benefits to BE Checker.

I've been able to find a handful of unclear/confusing formulations needing rephrasing. Some suggestions were offered in the attached, reviewed version of the manuscript.
Reviewer 4 Report
The manuscript proposed an in vitro methodology to assess BE between CTs and ODTs from the similarity of the permeation profiles evaluated in the BE Checker. The present work is logical and well planned. However, there are some issues to be fixed and I suggest a major revision with following comments:
1. Whether different drug excipients affect the accuracy of methodology? Please design experiments for demonstrating it.
2. The authors demonstrate the advantage and improvement of the BE Checker in this work compared with previous works in introduction. Please compare the experiment results in this work from previous results, and then prove this work of innovation.
3. In the manuscript, authors got different experimental results from 50 ppm and 100 ppm. It is proved that the rate of agitation paddle can simulate the gastric peristalsis capacity of patients. However, how to reflect the real conditions of different patients? Please design experiments to prove the universality for different patients of this methodology.
4. Please add photographs of the BE Checker.
5. Please state the reasons for choosing the pita vastatin and the memantine.
6. In the abstract, in order to read clarity, when you first stated the BE Checker, you should provide the full name of “BE” first. Please check the format and pay attention to the publication years of the references. There are too much too long sentences in whole manuscript, please shorten and simplify these sentences.
Please check the format and pay attention to the publication years of the references.
Reviewer 5 Report
An immense effort clearly went into the preparation of this work. We have some suggestions for improvement.
1. Figures 1 and 2 are too abstract. Place a physical picture next to the Schematic illustration can help readers understand better.
2. In Figure 4 A and C, y-axis is too high that the data is unclear. And for Figure 4A, how can the y-axis dissolved amount be 120%? Same for Figure5.
3. What’s your reason for choosing naftopidil, silodosin, pregabalin and imidafenacin as model drugs?
4. Minor punctuation and grammatical errors need to be resolved.
5. What’s your detailed experiment temperature?
Minor editing of English language required
Round 2
Reviewer 1 Report
The manuscript is OK for publishing now
Reviewer 4 Report
The manuscript with the title “Development of an In Vitro Methodology to Assess the Bioequivalence of Orally Disintegrating Tablets Taken Without Water” proposed an in vitro methodology to assess BE between CTs and ODTs from the similarity of the permeation profiles evaluated in the BE Checker. After carefully checking all the response to comments, I do not recommend accepting this article on the Pharmaceutics due to its lack of innovation and experiments that I mentioned before.
There are too much too long sentences in whole manuscript, please shorten and simplify these sentences.